# Antidiabetic Activity and Potential Mechanism of Amentoflavone in Diabetic Mice

**DOI:** 10.3390/molecules24112184

**Published:** 2019-06-11

**Authors:** Chengfu Su, Chuanbin Yang, Man Gong, Yingying Ke, Peipei Yuan, Xiaolan Wang, Min Li, Xiaoke Zheng, Weisheng Feng

**Affiliations:** 1School of Pharmacy, Henan University of Chinese Medicine, Zhengzhou 450046, China; suchengfu1988@163.com (C.S.); 715676034@163.com (M.G.); keyingying1988@sina.com (Y.K.); 15136153630@163.com (P.Y.); wxl_325@163.com (X.W.); 2Mr. and Mrs. Ko Chi Ming centre for Parkinson’s Disease Research, School of Chinese Medicine, Hong Kong Baptist University, Hong Kong 999077, China; yangchb@hkbu.edu.hk (C.Y.); limin@hkbu.edu.hk (M.L.)

**Keywords:** biochemical indexes, glucose metabolic enzyme, PI3K/Akt, GLUT4 translocation

## Abstract

Aim: To investigate the anti-diabetic activity of amentoflavone (AME) in diabetic mice, and to explore the potential mechanisms. Methods: Diabetic mice induced by high fat diet and streptozotocin were administered with amentoflavone for 8 weeks. Biochemical indexes were tested to evaluate its anti-diabetic effect. Hepatic steatosis, the histopathology change of the pancreas was evaluated. The activity of glucose metabolic enzymes, the expression of Akt and pAkt, and the glucose transporter type 4 (GLUT4) immunoreactivity were detected. Results: AME decreased the level of glucose, total cholesterol (TC), triglyceride (TG), low density lipoprotein cholesterol (LDL-C) and glucagon, and increased the levels of high density lipoprotein cholesterol (HDL-C) and insulin. Additionally, AME increased the activity of glucokinase (GCK), phosphofructokinase-1 (PFK-1), and pyruvate kinase (PK), and inhibited the activity of glycogen synthase kinase-3 (GSK-3), phosphoenolpyruvate carboxykinase (PEPCK), and glucose-6-phosphatase (G-6-Pase). Mechanistically, AME increased superoxide dismutase (SOD), decreased malondialdehyde (MDA), activation of several key signaling molecules including pAkt (Ser473), and increased the translocation to the sedimenting membranes of GLUT4 in skeletal muscle tissue. Conclusions: AME exerted anti-diabetic effects by regulating glucose and lipid metabolism, perhaps via anti-oxidant effects and activating the PI3K/Akt pathway. Our study provided novel insight into the role and underlying mechanisms of AME in diabetes.

## 1. Introduction

Diabetes mellitus is a complex, heterogeneous, and polygenic disease which is characterized by increasing circulating glucose concentrations associated with abnormalities in carbohydrate, protein and fat metabolism caused by the complete or relative insufficiency of insulin secretion and/or insulin action. According to the International Diabetes Federation, there will be around 693 million people suffering from diabetes in 2045 [1].

Treatment of diabetes mellitus involves diet control, exercise and the use of insulin and/or oral hypoglycemic drugs. However, many oral hypoglycemic agents have a number of serious adverse effects [2]. The adverse reactions limit the use of antidiabetic drugs. Therefore, looking for new antidiabetic drugs which have little adverse reaction appears to be very urgent. Traditional and natural indigenous medicines on diabetes prevention and treatment have become a research focus, and a large number of natural plants/plant products have been evaluated for their anti-diabetic effects [3].

*Selaginella tamariscina* (Beauv.) Spring is a traditional Chinese medicine, which is used in folk medicine for the treatment of various diseases including amenorrhea, dysmenorrhea, and chronic hepatitis. Our previous study has found that the total flavonoids extracted from *Selaginella tamariscina* (Beauv.) Spring has a good antidiabetic activity [4]. Furthermore, we found that amentoflavone was the main flavone of the total flavonoids of *Selaginella tamariscina* (Beauv.) Spring. Importantly, we also found that amentoflavone can improve insulin resistance in HepG2 cells [5], which may indicate that AME has anti-diabetic effects. However, whether the AME has hypoglycemic effects in animal models and the underlying mechanisms are not clear. Here, a model of diabetic mice induced by high fat diet and low dose streptozotocin were established. By using this model, we detected the hypoglycemic effect of AME and identified its potential mechanisms.

## 2. Results

### 2.1. Effects of AME on Body Weight and Fasting Blood Glucose (FBG) Levels in Diabetic Mice

As shown in Table 1, after the treatment for 8 weeks, treatment with RG, AMEI, AMEII, had no obvious change in body weight. The diabetic control group (DC) mice led to an over five-fold elevation of the blood glucose level compared with the normal control group (NC) mice (*p* < 0.01). Treatment with RG, AME II caused significant reduction (*p* < 0.05) in blood glucose level after 8 weeks. Treatment with AMEI also had an anti-hyperglycemic effect, although these did not reach statistical significance.

### 2.2. Effects of AME on Oral Glucose Tolerance Test in Diabetic Mice

Results of the oral glucose tolerance test conducted on diabetic mice fed with AME are shown in Table 2 in diabetic control mice, blood glucose reached the highest level at 30 min after oral glucose ingestion, and this hyperglycemia was maintained until 60 min, and then began to decrease. AMEII significantly prevented (*p* < 0.05) the increase in blood glucose levels after glucose administration at 120 min in comparison with the diabetic control group. AMEI could also decrease blood glucose levels at different time, although these did not reach statistical significance. RG had an obvious hypoglycemic effect at 0 min and 120 min compared with the diabetic group (*p* < 0.05).

### 2.3. Effects of AME on Insulin and Glucagon Levels in Diabetic Mice

Table 3 showed the levels of insulin and glucagon in serum in each group, it indicated that the level of insulin decreased significantly in diabetic control mice (*p* < 0.01) while the level of glucagon increased significantly (*p* < 0.01), whereas the RG, AMEI, AME II increased the insulin levels, and the RG and AME II showed significant changes when compared with the diabetic control mice (*p* < 0.01), moreover, the RG, AME II decreased the glucagon levels significantly (*p* < 0.01).

### 2.4. Effects of AME on Lipids and Lipoprotein in Diabetic Mice

As shown in Table 4, the serum levels of TG, TC, HDL-C and LDL-C were significantly raised (*p* < 0.05, *p* < 0.01) compared with normal control mice. After 8 weeks of treatment, the RG, AMEI, AME II could significantly lowered the levels of TG, LDL-C (*p* < 0.05, *p* < 0.01), while the RG, AMEI, AME II could significantly increase the HDL-C level in comparison to the diabetic controls (*p* < 0.01). Although the whole AME group could decrease the level of TC, only the AME II showed an obvious trend (*p* < 0.05), and RG could also decrease the level of TC significantly (*p* < 0.05).

### 2.5. Effects of AME on the Level of Alanine Aminotransferase (ALT), Aspartate Aminotransferase (AST), Blood Urea Nitrogen (BUN) and Creatinine (CREA) in Serum and Liver iNdex and Kidney Index in Diabetic Mice

As shown in Table 5 that the levels of ALT, AST and liver index had significantly increased in diabetic mice compared with normal mice (*p* < 0.01), and the AMEI, AMEII and RG significantly decreased the level of ALT, AST and liver index (*p* < 0.01) in comparison to diabetic control mice. The levels of BUN, CREA and kidney index of all the groups have no obvious change.

### 2.6. Effects of AME on Hepatic Steatosis

AS is shown in Figure 1, there is a high degree of steatosis with severe cytoplasmic vacuoles and swelling of hepatocytes of diabetic control mice. After being fed with AME, there is an obvious prevention of the fatty deposition in hepatocytes. Particularly, AMEII showed a greater improvement of the steatosis and had almost the same effect as RG.

### 2.7. Effects of AME on the Histopathology Change of the Pancreas

As was shown in Figure 2, the pancreatic sections of the DC group mice stained with HE showed the severe necrotic changes of pancreatic islets, especially in the center of the islets. Nuclear changes, karyolysis, disappearing of nucleus and in some places residue of destructed cells were visible. The relative reduction of the size and number of islets, especially around the large vessel, and a severe reduction of the number of B cells was obvious. Administration of AME and RG led to the protection of the histomorphologic change of the pancreas, increasing the number of the B cells. In particular, AMEII almost repaired the histomorphologic change of the pancreas.

### 2.8. Effects of AME on Antioxidant Parameters in Liver

Table 6 indicated that the MDA level significantly increased (*p* < 0.01) whereas the SOD activity significantly decreased (*p* < 0.01) in diabetic mice compared with normal control mice. AME II and RG significantly increased (*p* < 0.01) the SOD activity, AME II and RG significantly decreased (*p* < 0.01) the MDA level, while the AMEI had no obvious change compared with the diabetic control group.

### 2.9. Effects of AME on the Activity of Glucose Metabolic Enzymes in Liver of Diabetic Mice

Table 7 showed that the activity of GCK, PFK-1 and PK of the diabetic group decreased significantly (*p <* 0.05, *p <* 0.01), while the activity of GSK-3, PEPCK and G-6-Pase of the diabetic group increased obviously (*p <* 0.05, *p <* 0.01) compared with normal control mice. The AMEI, AME II and RG could significantly increase the activity of GCK, PFK-1 and PK (*p <* 0.05, *p <* 0.01), and decrease the activity of GSK-3, PEPCK and G-6-Pase (*p <* 0.05, *p <* 0.01) in comparison to diabetic control mice.

### 2.10. Effects of AME on Protein Expression of Akt Serine Phosphorylation (Ser-473)in Muscle of Diabetic Mice

As is shown in the Figure 3 and Figure 4. The pAkt(Ser-473) expression of the diabetic control mice significantly decreased 61% in diabetic control mice than that in the normal control mice, and the AME II significantly increased pAkt(Ser-473) expression by 69.23% above that of diabetic control mice (*p <* 0.01), and the RG group could increase the expression of pAkt(Ser-473) significantly (*p <* 0.01).

### 2.11. Effects of AME on GLUT4 Immunoreactivity in the Skeletal Muscle Tissue of Diabetic Mice

As shown in Figure 5 and Figure 6, the GLUT4 immunoreactivity in sections of skeletal muscle from all subjects expressed distinct granular reactions in association with the cell surface. Few grains were found in deeper parts of the cytoplasm. Due to its distinct granular appearance, the GLUT4 immunoreactivity could be quantified by counting. The GLUT4 integral optical density (IOD) of the diabetic control mice significantly decreased compared with that of normal control mice (*p <* 0.01), and the RG, AMEI, AME II could significantly increase the GLUT4 IOD (*p <* 0.01).

## 3. Discussion

Insulin resistance and insufficiency of insulin secretion are the basis of type 2 diabetes [6]. Insulin resistance is a condition where the body tissues become resistant to insulin, resulting in a marked decrease of glucose metabolism in response to insulin [7]; which is associated with dyslipidemia [8]. Studies have shown that the high fat diet feeding rats develop insulin resistance [9]. Meanwhile, streptozotocin has been known to selectively target and destroy the pancreatic β-cell by necrosis [10]. Our previous studies indicated that the total flavonoids of *Selaginella tamariscina* (Beauv.) Spring had hypoglycemic and hypolipidemic effects on the diabetic rats induced by high fat diet and low dose streptozotocin [4]. Therefore, we selected the mice model by high-fat diet following low-dose streptozotocin in the study to evaluate the effect of AME on the treatment of type 2 diabetes. Normally, diabetes is detected by measuring glucose blood levels. In addition, diabetic mice have impaired glucose tolerance. Additional load of glucose is found to impair the tolerance further, and an index of insulin sensitivity obtained from the oral glucose tolerance test (OGTT) [11]. Therefore, our research investigated FBG and OGTT during AME treatment. From the results obtained, AME at a dose of 40 mg/kg produced a statistically significant decrease in FBG in comparison to the diabetic control group. From the data obtained in the OGTT, it was clear that AME at a dose of 40 mg/kg blocked an increase in blood glucose levels after 120 min of glucose administration clearly, these results imply that AME has the ability to increase peripheral utilization of glucose, it has the same hypoglycemic effect as rosiglitazone.

The glucose metabolism is complex. At the molecular level, a variety of hormones can affect blood glucose levels. Insulin and glucagon play a prominent role in the regulation of glucose metabolism. They vary with the changes in the blood glucose level, and in turn regulate the blood glucose level precisely. Insulin is the unique hormone that down-regulate blood glucose level. In contrast to the action of insulin, glucagon is a hormone promoting the catabolism. It can promote glycogenolysis and gluconeogenesis, and increase the blood glucose level significantly. In the diabetic group, the insulin level was significantly reduced, while the level of glucagon was significantly increased. However, after the administration of AME, the level of insulin was significantly increased and the level of glucagon was significantly decreased. It implies that the regulation of insulin and glucagon may be one of the possible factors responsible for the anti-diabetic activity of AME in diabetic mice.

Diabetes is associated with profound alterations in the plasma lipid and lipoprotein profile. In uncontrolled type 2 diabetes mellitus, there will be an increase in TC, LDL-C, VLDL-C, TG and a decrease in HDL-C which contributes to the coronary artery disease [12]. In addition, hypertriglyceridemia is also an important maker of insulin resistance [13]. In our previous study, a rise in blood glucose was accompanied with the lipid metabolism disorder. In the present study, the rise in blood glucose was accompanied with a marked increase in TC, LDL-C, and TG in diabetic mice. AME could decrease the TC, TG, and LDL-C value while elevate the HDL-C value. It is implying that the AME could alter the disorder of lipid metabolism.

When there is damage in the liver, the level of ALT and AST in serum can increase [14]. BUN is the terminal product of protein metabolism; when kidney was damaged, the level of BUN in the blood increased. Serum creatinine concentration is widely used as an index of renal function [15]. Therefore, we detected the level of ALT, AST, BUN and CREA in serum to show the influence of AME on the liver and kidney function. In the present study, the level of ALT, AST and liver index was significantly increased in diabetic mice, it suggested that the liver was necrotized in diabetic mice. However, there was no obvious change on the BUN, CREA level and kidney index. AME could significantly decrease the level of ALT, AST and liver index; what’s more, AME didn’t change the level of BUN, CREA and kidney index. It implies that AME could effectively ameliorate hepatic injury in diabetic mice, and AME didn’t have renal injury.

Hepatic steatosis is an important marker of metabolic dysfunction, the type 2 diabetes mellitus is likely responsible for many cardiometabolic risk factors associated with hepatic steatosis [16]. In the study, AME ameliorated hepatic lipid accumulation of hepatic steatosis. The high dose of AME had an obvious effect.

In type 2 diabetes mellitus, defective insulin secretion is caused by beta cell dysfunction, and the number of the beta cell was reduced [17]. According to our study, the relative reduction of the size and number of islets, especially around the large vessel, and also severe reduction of the number of B cells was obvious in diabetic mice. AME could protect the histomorphologic change of the pancreas, adding the number of the B cells. In particular, the high dose of AME almost repaired the histomorphologic change of the pancreas.

The balance between the source and outlet of blood glucose keeps the blood glucose at the normal level. The blood glucose is from the digestive absorption of food, glycogenolysis and gluconeogenesis. The outlet of the blood is glycolysis, aerobic oxidation, glycogen synthesis, transforming into other sugars, lipid and amino acid et al. The whole process of the reaction is an enzymatic reaction. Some of enzymes in the reaction play an important role. In the process of glycolysis, hexokinase which is called glucokinase (GCK) in hepatocytes, phosphofructokinase-1(PFK-1) and pyruvate kinase (PK) are very important kinases. GCK can catalyze the phosphorylation of glucose to glucose 6-phosphate. The reduction of GCK activation may involve insulin resistance [18]. PFK-1 catalyzes the transfer of a phosphoryl group from ATP to fructose 6-phosphate to yield fructose 1,6-bisphosphate. PFK-1 reaction is essentially irreversible under cellular conditions, and it is the first “committed” step in the glycolytic pathway. Pyruvate kinase (PK) catalyzes the transfer of the phosphoryl group from phosphoenolpyruvate to ADP, the pyruvate kinase reaction is essentially irreversible under intracellular conditions and is an important site of regulation. The activity of PK plays a very important role in the level of blood sugar [19]. During the process of glycogen synthesis, glycogen synthase kinase-3(GSK-3) retains the activity of glycogen synthase by its phosphorylation, and then the glycogen synthesis is reduced. Restraining the activity of GSK-3 in this way can increase the activity of glycogen synthase, and increase the glycogen synthesis to decrease the level of blood sugar. GSK-3 can regulate the gene expression of phosphoenolpyruvate carboxykinase (PEPCK) and glucose-6-phosphatase (G-6-Pase) in the process of gluconeogenesis [20]. PEPCK is an important kinase in gluconeogenesis. G-6-Pase represents a second important gluconeogenic enzyme and also catalyzes the last step in glycogenolysis [21]. In addition, high fructose diet feeding increased expression of PEPCK and G-6-Pase [22]. In our study, in the diabetic group, the activity of GCK, PFK-1, PK was significantly reduced, while the activity of GSK-3, PEPCK, G-6-Pase was significantly increased compared with the normal group. However, after the administration of AME, we observed a significant increase in the activity of GCK, PFK-1, PK, and a significant decrease in the activity of GSK-3, PEPCK, and G-6-Pase. It is suggested that the pertinent anti-diabetic mechanism of AME may be related to promoting glycolysis, aerobic oxidation and glycogen synthesis, and to restraining the gluconeogenesis.

Now, studies have found that oxidative stress has become an important factor in the pathogenesis of diabetes [23]. SOD is the first enzyme of the scavenger enzyme series to protect tissues against oxygen free radicals by catalyzing the removal of superoxide radicals, which damage the membrane and biological structures [24]. MDA reflected the degree of lipid peroxidation and increased malonaldehyde production, which played an important role in the progression of diabetic pancreas damage [25]. Our results corroborated these observations. In the present study, an increase level of MDA and a decreased activity of SOD were noticed in diabetic mice. After the administration of AME for 8 weeks, a significant increase of SOD activities associated with decreased MDA levels in liver, indicating that antioxidative stress could be one of the mechanisms by which AME alleviated the high blood glucose status in diabetic mice.

The disorder of insulin signal transduction plays an important role in the pathogenesis of diabetes, so it has important significance with the study of insulin signal transduction [26], and the metabolic function of insulin is mainly mediated by the PI3K/Akt pathway [27]. The PI3K/Akt pathway is briefly introduced as follows. After insulin binding with an insulin receptor, the PI3K is activated, and then a series of biological effects is produced. In addition, the Akt is activated. Akt is an extremely important signaling protein in the pathway. Studies have shown that the phosphorylation of Akt (Ser473) is the primary phosphorylated form after the activation of Akt [28]. Generally, the phosphorylation level of Akt(Ser473) was used to reflect the activity of the PI3K/Akt pathway. The activation of Akt could obviously improve the impaired ability of GLUT4 translocation from intracellular storage vesicles to the cell membrane of the mice fed with a high fat diet. GLUT4 over expression in db/db mice ameliorates diabetes [29]. In our present study, the Akt protein of diabetic mice had no obvious change. However, the phosphorylation level of Akt(Ser473) of diabetic mice was decreased significantly while the protein of GLUT4 in skeletal muscle was decreased significantly. This suggests that activation of Akt is in disorder, and the effects of insulin cannot respond, leading to the disorder of the glucose metabolism. AME could markedly increase the level of phosphorylation Akt (Ser473) and increase the expression of GLUT4 in skeletal muscle of diabetic mice. This implies that AME may activate the PI3K/Akt pathway and promote GLUT4 transporting glucose to have a hypoglycemic effect.

In summary, this research revealed that amentoflavone, an active compound of *Selaginella tamariscina* (Beauv) Spring, had the effect of ameliorating the glucose, lipid metabolism disorder, and the hepatic lipid accumulation of hepatic steatosis, and repairing the histomorphologic change of pancreas. Moreover, AME had the effect of regulating the enzyme activity of glucose metabolism, increasing the insulin secretion and improving the insulin signal transduction in target tissues. These benefits may be via anti-oxidant effects and activating the PI3K/Akt pathway. However, evaluation of the hypoglycemic of amentoflavone in the complex mechanism needs to be further explored.

## 4. Materials and Methods

### 4.1. Preparation of AME

Dried *Selaginella tamariscina* (Beauv) Spring was purchased from the Ben Cao Guo Yao Tang (Zhengzhou, China) and appraised by Chen Sui-qing of the Henan University of Traditional Chinese Medicine. AME was isolated from *Selaginella.* The purity of the AME prepared in our laboratory is more than 98% as determined by assays of high performance liquid chromatography (HPLC) (Figure 7). All doses were given AME powder dissolved in distilled water according to the weight of the mice.

### 4.2. Chemicals and Reagents

CD-1(ICR) mice initially weighing 18–22 g were purchased from Vital River (Beijing, China) (Certificate no. SCXK (Jing) 2006–0009). Streptozotocin was purchased from Sigma Chemical Co. (St. Louis, MO, USA). Rosiglitazone was purchased from Taiji Group, Chongqing Fuling Pharmaceutical Factory (Chongqing, China). The kit for blood glucose was purchased from Biosino Bio-technology and Science Inc. (Beijing, China) The kits for total cholesterol (TC) and triglyceride (TG) were purchased from Zhejiang Audit Biotechnology Corp. (Zhejiang, China).The kits for high density lipoprotein-cholesterol (HDL-C), alanine aminotransferase (ALT), aspartate aminotransferase (AST), blood urea nitrogen (BUN) and creatinine (CREA) were purchased from Yantai Ausbio Biology Engineering Corp. (Shanghai, China). The kits for malondialdehyde (MDA) and superoxide dismutase (SOD) were purchased from Nanjingjiancheng Bioengineering Institute (Jiangsu, China). The kits for Insulin, and glucagon were purchased from Beijing North Institute of Biological Technology (Beijing, China). The enzyme-linked immunosorbent assay (ELISA) kit for glucosekinase (GCK), 6-phosphofructokinase-1 (PFK-1), pyruvate kinase (PK), glycogen synthase kinase-3 (GSK-3), phosphoenolpyruvate carboxykinase (PEPCK) and glucose-6-phosphatase (G-6-Pase) were all purchased from R&D Systems (Minneapolis, MN, USA) The total protein extraction kit and protease inhibitor were purchased from Beyotime Institute of Biotechnology (Jiangsu, China). Akt, pAkt, β-tubulin and GLUT4 rabbit poly-clonal antibodies were purchased from Abcam (Cambridge, UK). Anti-rabbit lgG HRP-conjugated antibody, eECL western kit and improved lowry protein assay kit were purchased from Cowin Biotech Co. (Beijing, China). Protein ladder was purchased from Fermentas Life Sciences (Burlington, ON, Canada). Polyvinylidene fluoride (PVDF) membranes were purchased from Millipore Corporation (Bedford, MA, USA). Broad spectrum SP immunohistochemical detection kit was purchased from Shanghai Yanhui Biological Technology Co.Ltd (Shanghai, China).The (Diaminobezidine) DAB kit was purchased from ZSGB-BIO company (Beijing, China). Organic solvents and other chemicals were of the highest analytical grade.

### 4.3. Experimental Model and Drug Treatment

Mice were maintained under standard laboratory conditions (temperature: 25 ± 2 °C, humidity: 60 ± 5%, 12 h dark/light cycle), and fed with a standard laboratory diet and fed a standard laboratory diet and water.

After a 1-week acclimation period, mice were randomly divided into two groups. The normal control group (10 mice) was fed with a basic diet, whereas the experimental group was fed with a high fat diet (consisting of 18% fat, 20% carbohydrate, 3% egg and 59% basic diet (*w*/*w*) which made by the Animal Experimental Center of Zhengzhou University) for a period of 8 weeks. After 8 weeks of dietary manipulation, the experimental mice were fasted overnight and were intraperitoneally injected with a freshly prepared solution of streptozotocin (100 mg/kg) in 0.1M citrate buffer (pH 4.21) to induce diabetic model, while the normal control mice were given the 0.1M citrate buffer in a dose volume of 1 mL/kg respectively. The mice with fasting plasma glucose level of above 11.1 mmol/L 72 h after injection of streptozotocin were considered diabetic and only uniformly diabetic mice were induced in the study.

The mice were divided into six groups: Group NC-normal control mice; Group DC-diabetic control mice; Group RG-diabetic mice treated with rosiglitazone (4 mg/kg, ig.); Group AMEI-diabetic mice treated with AME (20 mg/kg, ig.); Group AME II-diabetic mice treated with AME (40 mg/kg, ig.); the mice were treated for 8 weeks. Blood samples were collected from the mice fasted for 4 h previously every 4 weeks, and serum glucose levels were estimated. OGTT was performed the day before mice were sacrificed. At the end of the experiment, blood samples were collected from the eyes (venous pool) and centrifuged at 2900× *g* for 10 min to separate the plasma from the whole blood and stored at −80 °C until assayed. The hepatics and skeletal muscles were harvested, frozen in liquid N_2_, and subsequently stored at −80 °C until required.

All experimental animals were overseen and approved by the Institutional Animal Care and Use Committee of Henan University of Traditional Chinese Medicine before and during experiments.

### 4.4. Oral Glucose Tolerance Test (OGTT)

An oral glucose tolerance test (OGTT) was performed using an oral dose of glucose (2 g/kg) for 2 h after 8 consecutive weeks of AME or rosiglitazone treatment. Animals were food-restricted and were given only water to drink throughout the time prior to the OGTT procedure. The blood samples were collected from each group just before glucose administration (0 min) and at 30, 60 and 120 min after glucose administration. Plasma glucose concentrations were determined by the glucose oxidase method.

### 4.5. Biochemical Assays

Glucose levels were estimated by commercially available glucose kits based on the glucose oxidase method. TC, TG, HDL-C, LDL-C, ALT, AST, BUN and CREA levels were measured by using commercial assay kits according to the manufacture^r^’s directions. The liver index and kidney index were calculated as follows: liver index = the weight of the liver/the body weight, kidney index = the weight of the kidney/the body weight. The contents of MDA and the activity of SOD were determined by commercially available kits according to the manufacturer’s instructions.

### 4.6. Morphological Evaluation

The liver and the pancreas were stored in 10% formalin, and then the tissue was embedded in paraffin, sliced at 5 μm thickness, and dyed with hematoxylin and eosin (HE) staining. The pathological changes of the liver and the pancreas were assessed and photographed under an Olympus BX-51 microscope.

### 4.7. Enzyme-Linked Immunosorbent Assay (ELISA)

The liver homogenate was prepared, and then the activity of GCK, PFK-1, PK, GSK-3, PEPCK, G-6-Pase was analyzed by the enzyme-linked immunosorbent assay (ELISA) method according to the respective manufacturer’s instructions.

### 4.8. Western Blot Analysis

Protein was extracted from tissues following the commercial kits’ instructions. Protein concentration was determined by using the BCA protein assay kit. Equal amounts of protein samples were separated by sodium dodecyl sulphate polyacrylamide gel electrophoresis (SDS-PAGE), and transferred to the PVDF membrane. The membrane was blocked in 5% (*v*/*v*) non-fat milk in TBST (20 mM Tris-HCL, 137 mM NaCl, 0.1% (*v*/*v*) Tween-20) for 2 h at room temperature, and then blotted with a primary antibody (Akt at 1:500, pAkt at 1:2500) overnight at 4 °C. After being washed (6 × 5 min) in TBST buffer, the membrane was detected with the second anti-body (1:1000) for 1h at room temperature, followed by additional washes (6 × 5 min) in TBST. Bound antibody was visualized by enhanced chemiluminescence. The intensity of target proteins and reference protein were quantified using Gene tools. The mean value for samples from normal group mice on each immunoblot, expressed in densitometry units, was adjusted to a value of 1.0. All experimental sample values were then expressed relative to this adjusted mean value.

### 4.9. Immunohistochemical Assay

Skeletal muscle tissue samples were fixed for 48 h, embedded in paraffin, and then sliced. The thickness of the paraffin section was 5 μm. The section was dewaxed to water, then the 3% H_2_O_2_ was used to inactive endogenous enzymes for 10 min at room temperature, then being washed (3 × 3 min) in PBS (135 mM NaCl, 2.7 mM KCl, 1.5 mM KH_2_PO_4_, 8 mM Na_2_HPO_4_, pH 7.2). The section was immersed in the citrate buffer (pH 6.0), and heated by microwave, and then cooled, and washed (3 × 3 min) in PBS. The blocking solution I was added in the section for 20 min at room temperature, and then the excess liquid was shaken out. The section was incubated for 1 h with rabbit anti-GLUT4 antibody (1:500), followed by additional washes (3 × 3 min) in PBS. Then the biotinylated goat anti-rabbit antibody was added in the section for 50 min at 37 °C, and then being washed (3 × 3 min) in PBS; then the peroxidase complex was added in the section for 50 min at room temperature, followed by additional washes (3 × 3 min) in PBS. Then the section was colored by the DAB kit; and then the section was observed and photographed by electric microscope (OLYMPUS BX61). Moreover, the photos were analyzed by Image pro plus6.0. (3 paraffin sections were chosen for each group, and 5 photos were taken for each section.)

### 4.10. Statistical Analysis

Results are presented as mean ± SD, and the comparison between groups were performed by one-way ANOVA. *p* values < 0.05 were considered statistically significant.

## 5. Conclusions

AME exerted anti-diabetic effects by regulating glucose and lipid metabolism, possible via anti-oxidant effects and activating the PI3K/Akt pathway. Our study provided novel insight into the role and underlying mechanisms of AME in diabetes.

## Figures and Tables

**Figure 1 molecules-24-02184-f001:**
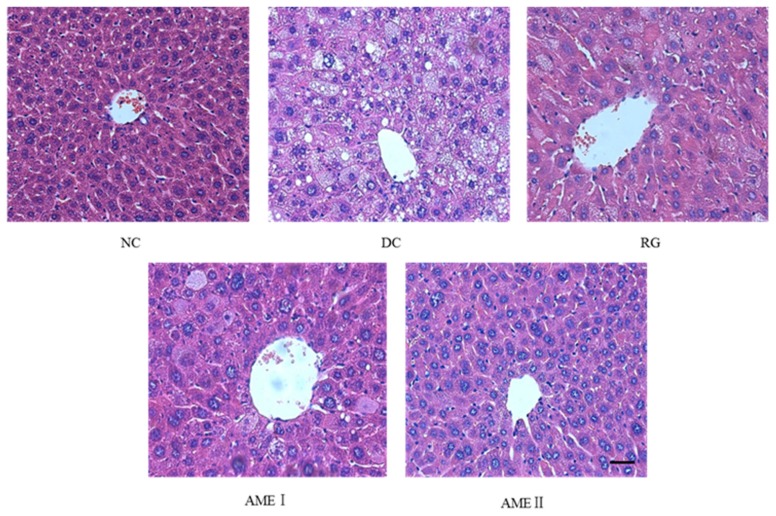
Effects of AME on hepatic steatosis in mice. The tissues were surgically excised and subjected to histological study by staining with hematoxylin and eosin. Magnification: 400×. The scale bar represents a length of 20 μm.

**Figure 2 molecules-24-02184-f002:**
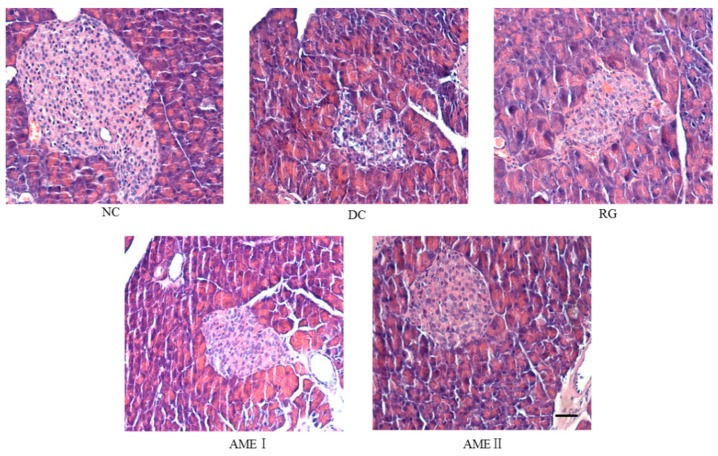
Effects of AME on the histopathology of the pancreas histomorphologic change of pancreas. The tissues were surgically excised and subjected to histological study by staining with hematoxylin and eosin. Magnification: 400×. The scale bar represents a length of 20 μm.

**Figure 3 molecules-24-02184-f003:**
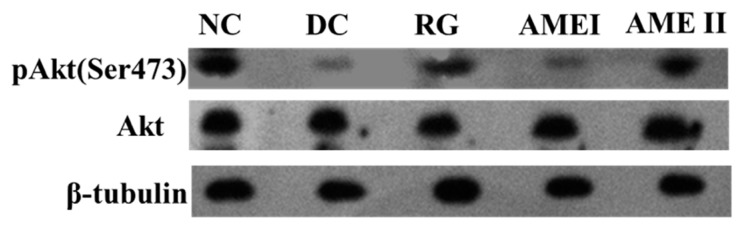
The effect of AME on pAkt, Akt, proteins (*n* = 10).

**Figure 4 molecules-24-02184-f004:**
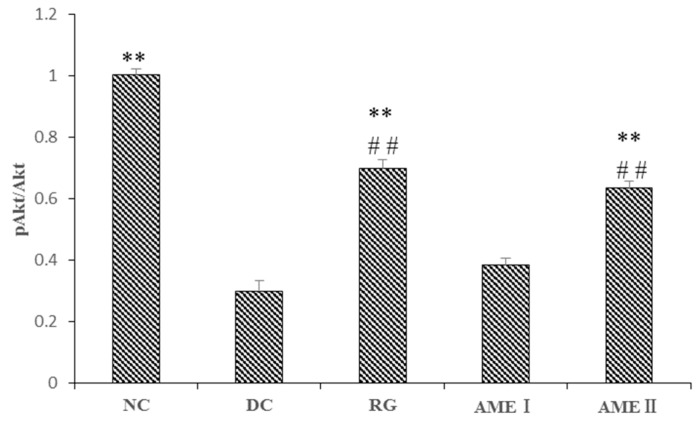
The effect of AME on pAkt, AKT proteins (*n* = 10), ^#^
*p* < 0.05, ^##^
*p <* 0.01 compared with normal control group; * *p <* 0.05, ** *p <* 0.01, compared with diabetic control group.

**Figure 5 molecules-24-02184-f005:**
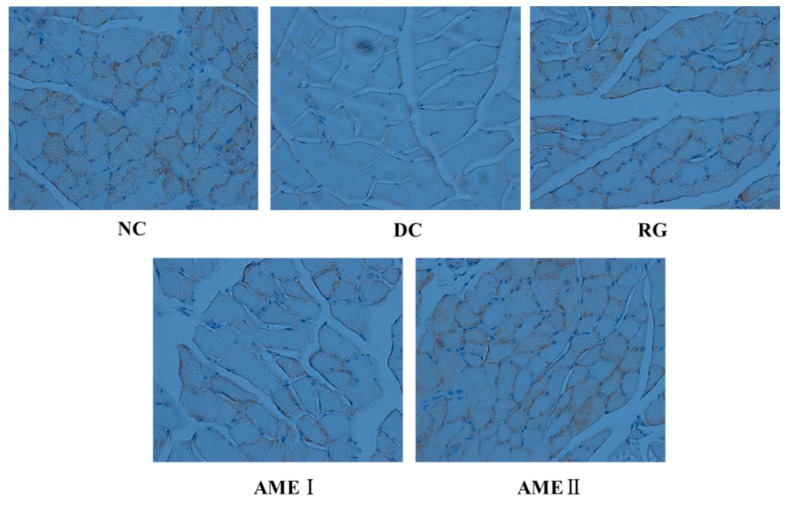
The effect of AME on GLUT4 immunoreactivity in the skeletal muscle tissue of diabetic mice (*n* = 10).

**Figure 6 molecules-24-02184-f006:**
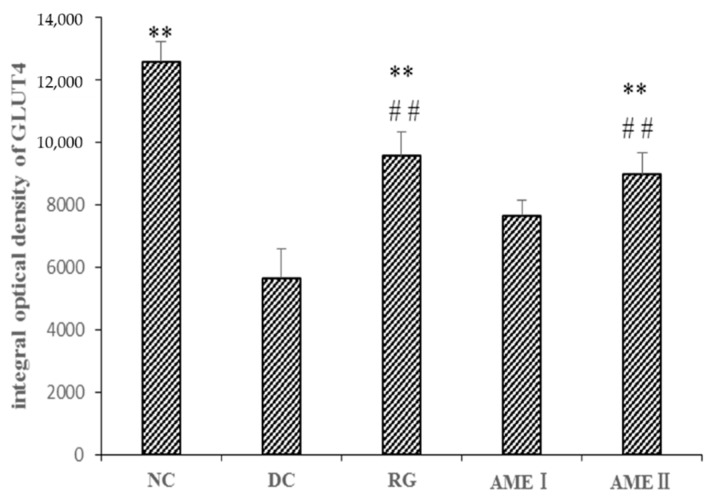
The effect of AME on GLUT4 immunoreactivity in the skeletal muscle tissue of diabetic mice. (*n* = 10), ^#^
*p* < 0.05, ^##^
*p <* 0.01 compared with normal control group; * *p <* 0.05, ** *p <* 0.01, compared with the diabetic control group.

**Figure 7 molecules-24-02184-f007:**
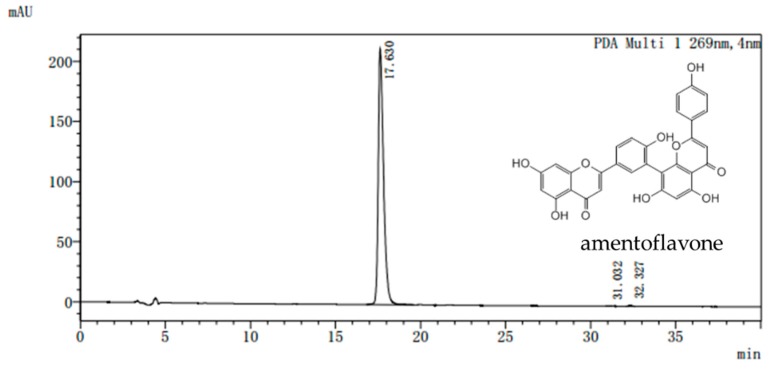
Representative high performance liquid chromatography (HPLC) chromatogram of amentoflavone. Waters Alliance 2695 separations module equipped with empower software hyphened with quaternary pumps, an automatic injector, a Waters 2998 photodiode array (PDA) detector at 190–800 nm, and a 250 mm × 4.6 mm × 5 μm Platisil ODS C18 column was used for separation. HPLC analysis conditions: 40% (*v*/*v*) acetonitrile /water run for 20 min, the conditions were maintained at 40% (*v*/*v*) acetonitrile /water and returned to 80% (*v*/*v*) acetonitrile /water from 20 to 40 min. flow velocity: 0.8 mL/min, wavelength of detection: 269 nm; column temperature: 40 °C.

**Table 1 molecules-24-02184-t001:** Effects of amentoflavone (AME) on body weight and Fasting Blood Glucose (FBG) levels in diabetic mice ($\bar{x}$ ± SD, *n* = 10).

Group	Body Weight (g/mouse)		FBG (mmol/L)	
	Before Treatment	After Treatment	Before Treatment	After Treatment
NC	41.60 ± 5 **	41.17 ± 5.33 **	3.25 ± 1.31 **	3.92 ± 1.17 **
DC	31.93 ± 3.79 ^##^	32.69 ± 3.37 ^##^	16.14 ± 4.15 ^##^	19.52 ± 2.42 ^##^
RG	32.21 ± 4.32 ^##^	33.16 ± 2.88 ^##^	16.14 ± 4.25 ^##^	17.80 ± 2.85 ^##^*
AMEI	32.79 ± 3.16 ^##^	34.38 ± 3.66 ^##^	15.98 ± 4.01 ^##^	18.05 ± 1.56 ^##^
AME II	34.23 ± 3.69 ^##^	33.59 ± 3.88 ^##^	15.91 ± 4.04 ^##^	17.59 ± 2.14 ^##^*

NC, normal control mice; DC, diabetic control mice; RG, diabetic mice treated with rosiglitazone (4 mg/kg, ig.), AMEⅠ, diabetic mice treated with AME (20 mg/kg, ig.); AMEII, diabetic mice treated with AME (40 mg/kg, ig.); ^#^
*p* < 0.05, ^##^
*p <* 0.01 compared with normal control group; * *p <* 0.05, ** *p <* 0.01, compared with diabetic control group.

**Table 2 molecules-24-02184-t002:** Effects of AME on oral glucose tolerance test in diabetic mice ($\bar{x}$ ± SD, *n* = 10).

Group	Blood Glucose Levels (mmol/L)			
	0 min	30 min	60 min	120 min
NC	3.92 ± 1.17 **	7.18 ± 1.54 **	5.08 ± 1.52 **	4.74 ± 1.58 **
DC	19.52 ± 2.42 ^##^	23.32 ± 2.64 ^##^	22.29 ± 2.19 ^##^	18.53 ± 1.41 ^##^
RG	17.80 ± 2.85 ^##*^	21.73 ± 2.80 ^##^	20.40 ± 2.99 ^##^	16.35 ± 2.65 ^##^*
AMEI	18.05 ± 1.56 ^##^	21.50 ± 3.25 ^##^	20.43 ± 3.55 ^##^	17.02 ± 2.19 ^##^
AME II	17.59 ± 2.14 ^##^*	21.69 ± 1.19 ^##^	20.87 ± 1.28 ^##^	16.56 ± 1.98 ^##^*

^#^*p* < 0.05, ^##^
*p <* 0.01 compared with normal control group; * *p <* 0.05, ** *p <* 0.01, compared with diabetic control group.

**Table 3 molecules-24-02184-t003:** Effects of AME on insulin and glucagon levels in diabetic mice ($\bar{x}$ ± SD, *n* = 10).

Group	Insulin (μIU/mL)	Glucagon (pg/mL)
NC	24.25 ± 4.89 **	154.85 ± 22.63 **
DC	16.12 ± 2.97 ^##^	254.54 ± 40.04 ^##^
RG	19.93 ± 2.38 ^#^	212.29 ± 36.72 ^##^**
AMEI	19.91 ± 3.90 ^#^	223.09 ± 31.27 ^##^
AME II	21.39 ± 3.62 **	209.87 ± 34.89 ^##^**

^#^*p* < 0.05, ^##^
*p <* 0.01 compared with normal control group; * *p* < 0.05, ** *p* < 0.01, compared with diabetic control group.

**Table 4 molecules-24-02184-t004:** Effects of AME on lipids and lipoprotein in diabetic mice ($\bar{x}$ ± SD, *n* = 10).

Group	TC (mmol/L)	TG (mmol/L)	HDL-C (mmol/L)	LDL-C (mol/L)
NC	2.80 ± 0.39 **	1.09 ± 0.33 **	2.18 ± 0.38 *	0.64 ± 0.09 **
DC	7.20 ± 1.27 ^##^	2.95 ± 1.18 ^##^	2.97 ± 0.22 ^#^	3.54 ± 1.37 ^##^
RG	6.34 ± 0.98 ^##^*	1.38 ± 0.47 **	4.47 ± 1.03 ^##^**	2.36 ± 0.36 ^##^**
AMEI	6.69 ± 1.01 ^##^	1.31 ± 0.38 **	4.27 ± 0.33 ^##^**	2.58 ± 0.90 ^##^*
AME II	6.60 ± 0.80 ^##^*	1.33 ± 0.63 **	4.39 ± 0.50 ^##^**	2.48 ± 0.81 ^##^**

^#^*p* < 0.05, ^##^
*p <* 0.01 compared with normal control group; * *p* < 0.05, ** *p* < 0.01, compared with diabetic control group.

**Table 5 molecules-24-02184-t005:** Effects of AME on the level of ALT, AST, BUN and CREA in serum and liver index and kidney index in diabetic mice ($\bar{x}$ ± SD, *n* = 10).

Group	ALT (U/L)	AST (U/L)	BUN (mmol/L)	CREA (μmol/L)	Liver Index (mg/g)	Kidney Index (mg/g)
NC	51.44 ± 6.39 **	134.75 ± 13.90 **	6.72 ± 1.15	77.98 ± 9.11	44.75 ± 2.17 **	16.66 ± 1.08
DC	384.33 ± 42.96 ^##^	530.38 ± 39.13 ^##^	6.79 ± 1.16	72.77 ± 10.96	91.73 ± 11.67 ^##^	17.44 ± 2.54
RG	308.17 ± 48.82 ^##^**	369.80 ± 14.73 ^##^**	6.31 ± 0.43	75.08 ± 8.69	67.34 ± 8.39 ^##^**	17.02 ± 1.74
AME I	297.57 ± 54.43 ^##^**	458.26 ± 30.90 ^##^**	6.69 ± 0.76	81.27 ± 9.36	63.61 ± 11.76 ^##^**	17.01 ± 2.61
AME II	222.00 ± 36.78 ^##^**	301.92 ± 37.59 ^##^**	6.69 ± 1.21	81.59 ± 7.83	70.59 ± 7.53 ^##^**	18.19 ± 1.83

^#^*p* < 0.05, ^##^
*p <* 0.01 compared with normal control group; * *p <* 0.05, ** *p <* 0.01, compared with diabetic control group.

**Table 6 molecules-24-02184-t006:** Effects of AME on malondialdehyde (MDA) level and superoxide dismutase (SOD) activity in diabetic mice ($\bar{x}$ ± SD, *n* = 10).

Group	SOD (U/mg)	MDA (nmol/mg)
NC	213.48 ± 33.49 **	4.30 ± 1.57 **
DC	162.14 ± 17.33 ^##^	9.58 ± 3.13 ^##^
RG	232.48 ± 20.69 **	6.58 ± 1.55 ^#^**
AMEI	169.22 ± 29.94	8.13 ± 1.74 ^##^
AME II	207.97 ± 26.59 **	4.39 ± 1.13 **

^#^*p* < 0.05, ^##^
*p <* 0.01 compared with normal control group; * *p* < 0.05, ** *p* < 0.01, compared with diabetic control group.

**Table 7 molecules-24-02184-t007:** Effects of AME on the activity of glucose metabolic enzymes in liver of diabetic mice ($\bar{x}$ ± SD, *n* = 10).

Group	GCK (u/mg)	PFK-1 (u/mg)	PK (mU/mg)	GSK-3 (pmol/mg)	PEPCK (IU/mg)	G-6-Pase (mIU/mg)
NC	1.37 ± 0.26 **	39.03 ± 2.18 **	68.66 ± 5.49 **	43.94 ± 7.67 **	4.45 ± 0.60 **	190.76 ± 19.33 **
DC	0.56 ± 0.10 ^##^	19.16 ± 4.47 ^##^	39.55 ± 5.59 ^##^	62.92 ± 4.89 ^##^	6.45 ± 0.54 ^##^	278.94 ± 14.28 ^##^
RG	1.00 ± 0.28 ^#^**	27.56 ± 2.50 ^##^**	59.38 ± 11.18 **	45.13 ± 6.37 **	4.64 ± 1.03 **	216.40 ± 25.62 **
AMEI	0.60 ± 0.08 ^##^	24.84 ± 4.42 ^##^*	57.13 ± 9.45 ^#^**	55.76 ± 10.39 ^#^*	4.90 ± 1.28 **	237.64 ± 3.24 ^##^*
AME II	1.11 ± 0.23 **	30.61 ± 4.39 ^##^**	57.07 ± 5.46 ^##^**	46.41 ± 8.12 **	4.07 ± 0.63 **	211.28 ± 22.81 *

^#^*p* < 0.05, ^##^
*p <* 0.01 compared with normal control group; * *p <* 0.05, ** *p <* 0.01, compared with diabetic control group.

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
