# Peer review of "Antidiabetic Activity and Potential Mechanism of Amentoflavone in Diabetic Mice"

_molecules, 2019, doi:10.3390/molecules24112184_

Reviewer 1 Report

Su and collaborators present a manuscript investigating, in a mouse models, the potential beneficial metabolic effects of and antidiabetic activity of the polyphenol amentoflavone.

Amentoflavone, derived from an herb used in traditional chinease medicine, indeed alleviates experimentally induced diabetes in mice. The manuscript is interesting, and built upon a lot of experimental evidence. I have, however, some major concerns relatively to the presentation of immunohistochemistry figures and western blotting figures.

Figure 1: there is no scale bar and micrographs seem to have been taken at different magnification. It is impossible to compare the data in this circumstance.

Western blot data (and the relative quantifications) cannot be presented as such. Measuring p85 does not provide any information on the activation status of PI3K. Likewise, measuring of PKB should be done in starving and re-fed (or insulin stimulated) animals.

GLUT4 immunihistochemistry is questionable: why are the micrographs presenting a blue background?

Other minor points:

Abstract: the statement  “the phosphorylation 3-kinase (PI3K)/Akt expression” makes no sense, please explain

Table 1 caption: please specify the nature of each experimental group

Author Response

Point 1: Figure 1: there is no scale bar and micrographs seem to have been taken at different magnification. It is impossible to compare the data in this circumstance.

Response 1: The new picture has been replaced, the scale bar represents a length of 20mm, and the scale bar is added in the micrographs.

Point 2: Western blot data (and the relative quantifications) cannot be presented as such. Measuring p85 does not provide any information on the activation status of PI3K. Likewise, measuring of PKB should be done in starving and re-fed (or insulin stimulated) animals.

Response 2: According to the Reviewer’s comments, we remove the band of p85 regulatory subunit of PI3K, and delete the description of the protein on our original paper. We don’t do the work of measuring of PKB in starving and re-fed (or insulin stimulated) animals. Future experiments have been designed and related results will be published in the future if it works.

Point 3: GLUT4 immunihistochemistry is questionable: why are the micrographs presenting a blue background?

Response 3: The reason we select the blue background is mainly according to the reference: Arluison, M.; Quignon, M.; Nguyen, P.; Thorens, B.; Leloup, C.; Penicaud, L., Distribution and anatomical localization of the glucose transporter 2 (GLUT2) in the adult rat brain--an immunohistochemical study. J Chem Neuroanat 2004, 28, (3), 117-36. In some glucose transporter micrographs of the paper, the background is blue, we think it is clearer to view the positive pots of the glucose transporter.

Point 4: Abstract: the statement “the phosphorylation 3-kinase (PI3K)/Akt expression” makes no sense, please explain.

Response 4: We have made correction according to the Reviewer’s comments. We change “the phosphorylation 3-kinase (PI3K)/Akt expression” into “the expression of Akt and pAkt”, and we had made correction on our original manuscript.

Point 5: Table 1 caption: please specify the nature of each experimental group.

Response 5: According to the Reviewer’s comments, we have added the content: “NC, normal control mice; DC, diabetic control mice; RG, diabetic mice treated with rosiglitazone (4mg/kg, ig.), AME, diabetic mice treated with AME (20mg/kg, ig.); AME, diabetic mice treated with AME (40mg/kg, ig.);” below the table 1 on our original manuscript.

Reviewer 2 Report

Here Su et al investigate the potential antidiabetic (hypoglycemic and hypolipidemic) effects of naturally occurring molecules, amentoflavone-I and II, in a mouse model of poorly-controlled type 2 diabetes. The authors associate these effects with increased activity of enzymes that promote glucose utilization/decreased activity of enzymes that promote glucose production, as well as – likely more importantly – increased activity of the insulin signaling cascade, including Akt phosphorylation and GLUT4 immunoreactivity. They use appropriate caution in describing their mechanistic findings, because they have not proven – only implied – that these mechanisms mediate AME’s effects. This is a well-done study addressing an important topic, and I am largely convinced by the authors’ findings. I have three minor concerns:

1.     The manuscript needs careful English review. Many sentences are clumsy and confusing.

2.     “Liver index” and “kidney index” are not commonly known parameters – please define in the methods how these were calculated.

3.     The authors state that the purity of AME was >98%. This is very important; please show these data in the manuscript.

Author Response

Point 1: The manuscript needs careful English review. Many sentences are clumsy and confusing.

Response 1: The manuscript has been edited by the MDPI providing English editing website.

Point 2: “Liver index” and “kidney index” are not commonly known parameters – please define in the methods how these were calculated.

Response 2: The liver index and kidney index were calculated like this: liver index=the weight of the liver/the body weight, kidney index=the weight of the kindey/the body weight. We add it in our original paper in line: 396-398.

Point 3: The authors state that the purity of AME was >98%. This is very important; please show these data in the manuscript.

Response 3: According the Reviewer’s comments, we add the Representative HPLC chromatogram of amentoflavone (Figure 7.) on our original paper.

Round  2

Reviewer 1 Report

No further comments apply.